# *Vestigial* mediates the effect of insulin signaling pathway on wing-morph switching in planthoppers

Jin-Li Zhang[1], Sheng-Jie Fu[1], Sun-Jie Chen[1], Hao-Hao Chen[1], Yi-Lai Liu[1], Xin-Yang Liu[1], Hai-Jun Xu[1,2,3]*

**1** Institute of Insect Sciences, Zhejiang University, Hangzhou, China, **2** State Key laboratory of Rice Biology, Zhejiang University, Hangzhou, China, **3** Ministry of Agriculture Key laboratory of Molecular Biology of Crop Pathogens and Insect Pests, Zhejiang University, Hangzhou, China

* haijunxu@zju.edu.cn

## Abstract

Wing polymorphism is an evolutionary feature found in a wide variety of insects, which offers a model system for studying the evolutionary significance of dispersal. In the wing-dimorphic planthopper *Nilaparvata lugens*, the insulin/insulin-like growth factor signaling (IIS) pathway acts as a 'master signal' that directs the development of either long-winged (LW) or short-winged (SW) morphs via regulation of the activity of Forkhead transcription factor subgroup O (*Nl*FoxO). However, downstream effectors of the IIS–FoxO signaling cascade that mediate alternative wing morphs are unclear. Here we found that *vestigial* (*Nlvg*), a key wing-patterning gene, is selectively and temporally regulated by the IIS–FoxO signaling cascade during the wing-morph decision stage (fifth-instar stage). RNA interference (RNAi)-mediated silencing of *Nlfoxo* increase *Nlvg* expression in the fifth-instar stage (the last nymphal stage), thereby inducing LW development. Conversely, silencing of *Nlvg* can antagonize the effects of IIS activity on LW development, redirecting wing commitment from LW to the morph with intermediate wing size. In vitro and in vivo binding assays indicated that *Nl*FoxO protein may suppress *Nlvg* expression by directly binding to the first intron region of the *Nlvg* locus. Our findings provide a first glimpse of the link connecting the IIS pathway to the wing-patterning network on the developmental plasticity of wings in insects, and help us understanding how phenotypic diversity is generated by the modification of a common set of pattern elements.

## Author summary

Many insects are capable of developing into either long-winged or short-winged adults, but the underlying molecular basis remains largely unknown. Pioneer studies showed that the insulin/insulin-like growth factor signaling pathway acts as a 'master signal' that directs wing buds to develop into long or short wings in the wing-dimorphic planthopper, *Nilaparvata lugens*. However, downstream effectors mediating the IIS pathway effects are unknown. Our findings highlight that *vestigial*, a key wing-patterning gene, is a main

**Data Availability Statement:** The raw data from the RNA-seq were submitted to GenBank (SRA accession number: PRJNA639300). The cDNA sequences of NlVg and SfVg were deposited in

GenBank with accession numbers of MT552978 and MT552979, respectively.

**Funding:** HJX was supported by grants from National Natural Science Foundation of China (http://www.nsfc.gov.cn/) (Grants 31522047, 31772158, and 31972261). The funders have no role in the study design, data collection and analysis, decidson to publish, or preparation of the manuscript.

**Competing interests:** The authors declare that they have no competing interest.

downstream effector that mediates the IIS activity on the development of alternative wing morphs during the wing-morph decision stage. The molecular mechanism of wing formation, including the function of *vestigial*, has been studied in great depth in the model insect *Drosophila melanogaster*. Our data provide a first glimpse of the link connecting the IIS pathway to the wing-patterning network in regulating developmental plasticity of wings in insects.

## Introduction

Wing polymorphism in insects offers an attractive model system for studying the evolutionary significance of dispersal [1,2]. In a variety of species, juvenile insects have an option to develop into either long-winged (LW) or short-winged (SW) (or wingless) adults caused by environment cues encountered during particular juvenile stages or by different genotypes, or by a combination of both [3–6]. The LW morph has fully developed wings and functional flight muscles, and is thus capable of flight, escaping from deteriorating environments and colonizing new habitats. By contrast, the SW or wingless morph has underdeveloped wings and flight muscles, and is thus obligately flightless. However, the SW or wingless morph may outcompete the LW morph via compensating alternative life-history traits such as reproduction [1,6]. Despite the ecological and evolutionary significance of wing polymorphism in numerous insects, the molecular basis underlying the developmental plasticity of wings is not well understood.

The brown planthopper (BPH), *Nilaparvata lugens* (Hemiptera: Delphacidae), has distinct LW and SW adults that are easily recognized from their external morphology (Fig 1A), and thus has long served as an example for understanding the mechanisms of wing polymorphism in insects [6–8]. As a direct-developing insect that lacks a pupal stage, BPHs progress through five nymphal instars (3–5 days for each stadium) and then molt into the adult form. BPH wing buds grow gradually with increasing nymph stage, but the SW and LW morphs are externally indistinguishable until the adult emerges. Previous studies revealed that the insulin/insulin-like growth factor signaling (IIS) pathway directs wing-morph switching via regulation of the activity of Forkhead transcription factor subgroup O (*Nl*FoxO) in the planthopper family [9–11] (Fig 1B). RNA interference (RNAi)-mediated gene silencing of one insulin receptor gene (*NlInR2*) increases the activity of the other receptor (*Nl*InR1), which in turn suppresses *Nl*FoxO activity and transforms wing commitment from SW to LW [10]. In contrast, RNAi-mediated silencing of *NlInR1* activates *Nl*FoxO, which subsequently inhibits LW development. Thus, silencing of *NlInR2* or *Nlfoxo* leads to LW morphs; whereas, silencing of *NlInR1* leads to SW morphs (Fig 1B). It was recently found that the IIS pathway is also involved in wing polymorphism control in the red-shouldered soapberry bug *Jadera haematoloma* (Hemiptera: Rhopalidae) [12] and the linden bug *Pyrrhocoris apterus* (Hemiptera: Pyrrhocoridae) [13], indicating that IIS has an evolutionarily conserved role on the developmental plasticity of wing buds in Hemiptera insects. However, relatively little is known about downstream effectors of the IIS–FoxO signaling cascade that are involved in wing-morph switching.

Here we report that the expression of *Nlvg*, a key component of the wing-patterning network in insects, is selectively and temporally regulated by the IIS–FoxO signaling cascade during the wing-morph decision stage. *Nl*FoxO protein may suppress *Nlvg* expression by directly binding to the first intron region of the *Nlvg* locus, and thus *Nlfoxo* dysfunction elevates *Nlvg* expression to induce LW development. Our findings provide new insight into the crosstalk

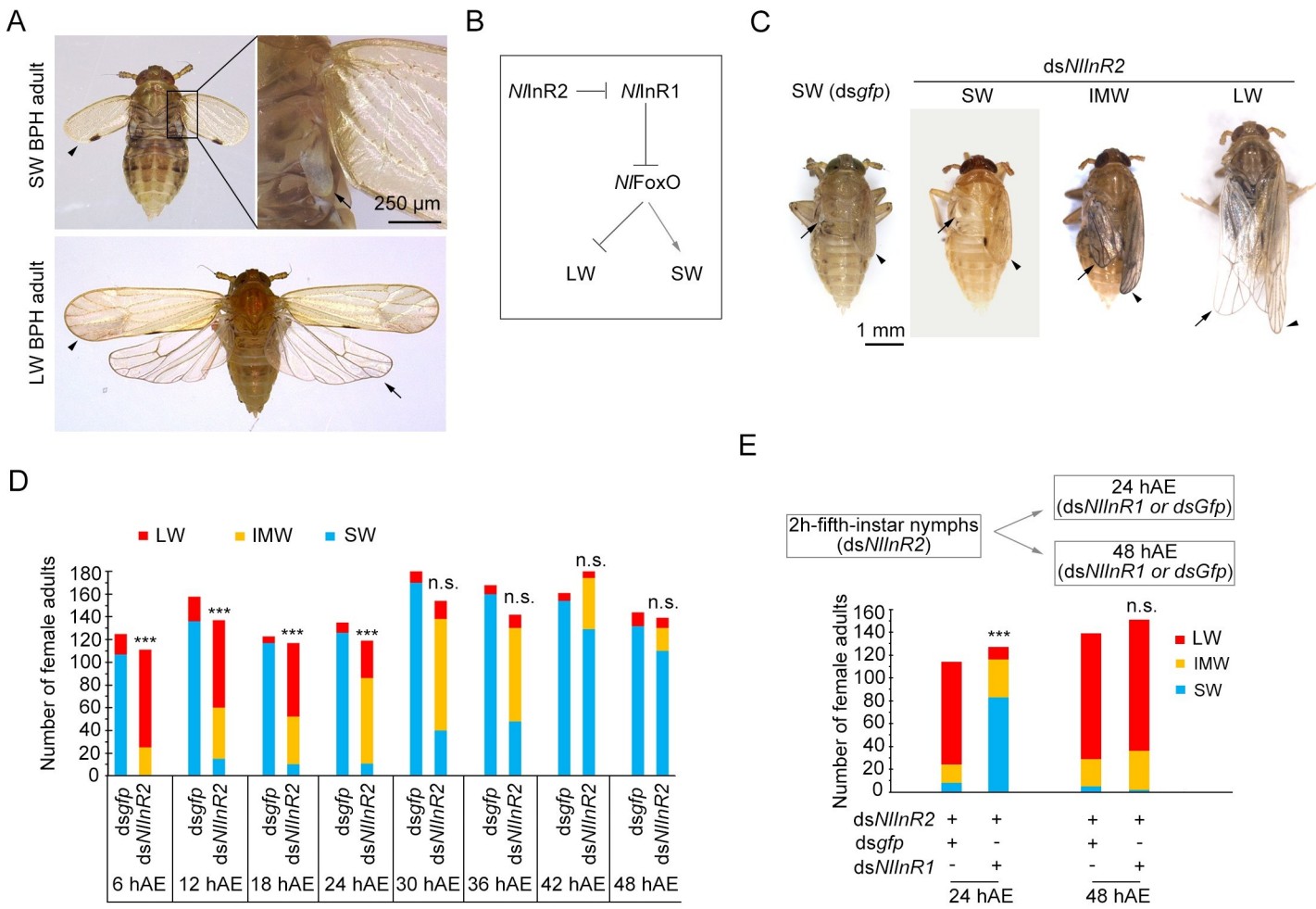

**Fig 1. Wing-morph switching during the wing-morph decision stage. (A)** Wild-type LW and SW BPH adults. **(B)** Schematic diagram of regulation of *Nl*InR1, *Nl*InR2, and *Nl*FoxO on LW and SW BPHs. *Nl*InR1 and *Nl*InR2 have opposite roles in wing-morph development, and depletion of *Nl*InR1 and *Nl*InR2 leads to SW and LW, respectively. *Nl*FoxO negatively regulates LW development, and depletion of *Nl*foxo leads to LW morphs. **(C)** Female adults with different wing morphs previously treated with ds*Nl*InR2 or ds*gfp*. Forewings at the left side were removed. **(D)** Numbers of female adults with different wing morphs after ds*Nl*InR2 treatment. Fifth-instar nymphs collected at designed time (6, 12, 18, 24, 30, 36, 42, and 48 hAE) were microinjected with ds*Nl*InR2 or ds*gfp*. hAE, hours after ecdysis. **(E)** Numbers of female adults with different wing morphs after double-gene knockdown. Fifth-instar nymphs at 2 hAE were microinjected with ds*Nl*InR2, and then microinjected with ds*Nl*InR1 or ds*gfp* at 24 and 48 hAE. SW, short-winged. IMW, intermediate-size wings. LW, long-winged. Arrowheads, forewings. Arrows, hindwings. Non-significant (n.s.) and significant (***$P < 0.001$, Pearson's $\chi^2$ test) differences from the control group (ds*gfp*) are indicated.

between the IIS pathway and wing-patterning network on regulation of developmental plasticity of wings.

## Results

### The early stage of the fifth instar is the binary decision period for SW and LW morphs

To unravel the downstream effectors that mediate the IIS pathway in the regulation of wing dimorphism in BPHs, determination of the decision-making period for wing-morph switching becomes an essential prerequisite. For this purpose, fifth-instar female nymphs from a wild-type (*wt*) SW-BPH strain (SW ratio > 90%) were collected at 6 h intervals after ecdysis, and then microinjected with double-stranded RNAs (dsRNAs) targeting *NlInR2* (ds*NlInR2*) or the

gene encoding green fluorescence protein (ds*gfp*), representing LW- and SW-destined morphs [10], respectively. In the ds*NlInR2* treatment group, the majority of fifth-instar nymphs at 6 h after ecdysis (hAE) developed into LW adults (Fig 1C and 1D), in stark contrast to SW adults in the ds*gfp*-treated group (Pearson $\chi^2$ test: $\chi^2$ = 94.904, *df* = 1, *P* < 0.001). Notably, owing to a mild RNAi effect, ds*NlInR2* treatment resulted in a small fraction of adults with intermediate-size wings (IMW; Fig 1C and 1D), with forewings and hindwings apparently smaller than those of LW adults (Fig 1C). Of particular note, the IMW phenotype was not observed when third- or fourth-instar nymphs were subjected to ds*NlInR2* treatment, which resulted in LW adults only [10]. By contrast, ds*gfp* treatment had no significant effect (Fig 1C), and generated as high a proportion of SW adults as *wt* SW BPHs (Fig 1D). The ds*NlInR2* effect persisted from 6 to 24 hAE fifth-instar nymphs (Pearson $\chi^2$ test: $\chi^2$ = 20.334, *df* = 1, *P* < 0.001), during which the relative proportion of LW adults was significantly greater than in the ds*gfp* treatment group. As development proceeded from 24 to 48 hAE, the ds*NlInR2* effect gradually faded, and the majority of nymphs were destined to develop into SW adults, although some IMW adults still developed (Fig 1D). A similar phenomenon was also observed for male nymphs, except that the ds*NlInR2* effect persisted until 36 hAE (Pearson $\chi^2$ test: $\chi^2$ = 6.556, *df* = 1, *P* = 0.01; S1 Fig). Thus, these data demonstrate unequivocally that the early stage of the fifth instar is a decision-making period for wing-morph switching from SW to LW in both female and male nymphs.

We next investigated whether the early stage of the fifth instar was also a decision-making period for LW to SW transition. Given that silencing of *NlInR1* could antagonize the ds*NlInR2* effect and thus redirect wing development from LW to SW morphs [10] (Fig 1B), we conducted a double-gene RNAi assay (ds*NlInR2*;ds*NlInR1*) in fifth-instar nymphs. The ds*NlInR2*; ds*NlInR1* treatment was capable of redirecting wing development from LW to SW in fifth instars at 24 hAE (Pearson $\chi^2$ test: $\chi^2$ = 121.903, *df* = 1, *P* < 0.001) but not at 48 hAE (Pearson $\chi^2$ test: $\chi^2$ = 0.718, *df* = 1, *P* = 0.397) (Fig 1E). These results indicate that the developmental trajectory of wing buds is reversible before the decision point for wing-morph switching; otherwise, wing buds are destined to form either LW or SW morphs when the decision period has passed. Taken together, these findings suggest that the decision-making period for wing-morph switching in BPH is confined to the early stage of the fifth instar.

## *Nlvg* is temporally regulated by fluctuating levels of the IIS activity during the fifth instar stage

To investigate further the mechanism by which *Nl*InR2 induces and executes LW development, we performed RNA sequencing (RNA-seq) for wing buds treated with either ds*NlInR2* or ds*gfp*. Nymphs at the onset of the fifth instar (2 hAE) were microinjected with ds*NlInR2* and ds*gfp* to generate LW- and SW-destined BPHs, respectively. Nota were dissected from 24- and 48-hAE fifth-instar nymphs for comparative transcriptomic analysis (Fig 2A). The results showed that ds*NlInR2* had a profound impact on genome-wide gene expression at 48 hAE but not at 24 hAE (S1 Data and S2–S5 Tables). At 48 hAE, compared to ds*gfp* treatment, ds*NlInR2* treatment significantly down-regulated 223 genes (adjusted *P* < 0.05) and concomitantly up-regulated 76 genes (Fig 2B). Of note, the *N. lugens* wing-patterning gene *vestigial* ortholog (*Nlvg*) was significantly up-regulated in ds*NlInR2*-treated nymphs (Fig 2B). The product of *vestigial* is a nuclear protein, and earlier studies showed that the *Drosophila vestigial* ortholog (*dvg*) is critical for wing development and patterning [14–18]. To confirm the transcriptomic data, we determined *Nlvg* expression levels in the context of *NlInR2* knockdown via quantitative real-time PCR (qRT-PCR). Third-instar nymphs were microinjected with *dsNlInR2*, and *Nlvg* transcripts were then measured at the developmental time indicated. These results

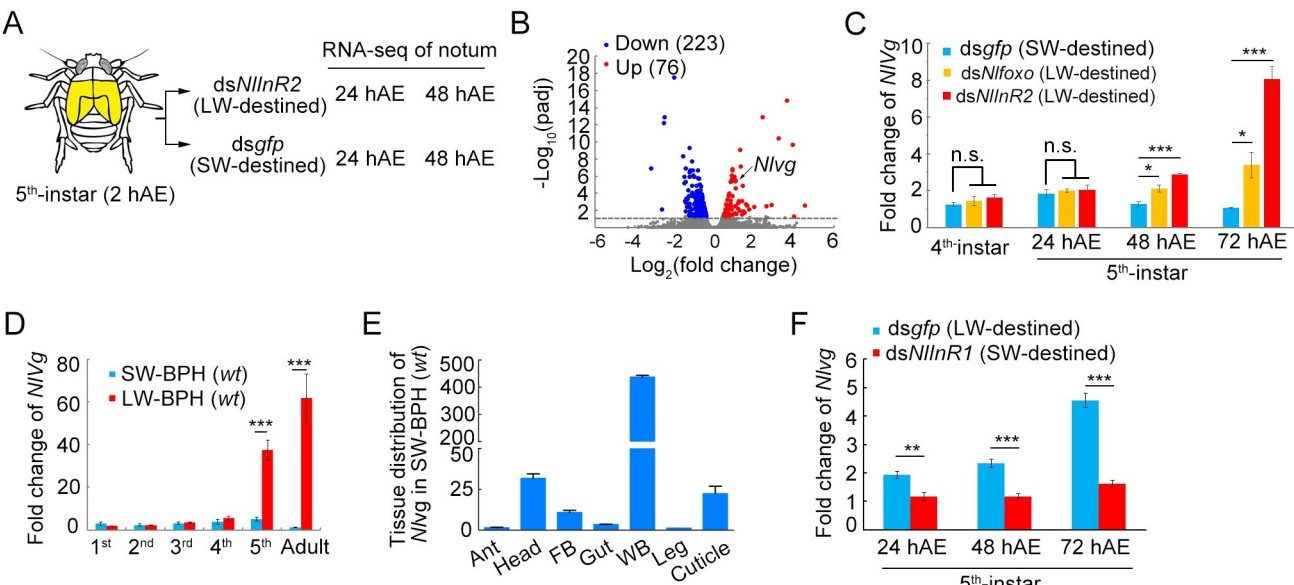

**Fig 2. Regulation of *Nlvg* by the IIS pathway. (A)** Schematic depiction of the notum tissue used for RNA-seq. Fifth-instar nymphs at 2 h after ecdysis (hAE) were microinjected with ds*NlInR2* or ds*gfp*. At 24- and 48-hAE, notum (indicate in yellow) was dissected for RNA-seq. **(B)** Numbers of up-regulated and down-regulated genes in ds*NlInR2*-treated notum compared to ds*gfp* treatment. The *Nlvg* expression level is indicated by an arrow. **(C)** Quantitative real-time PCR (qRT-PCR) analysis of *Nlvg* expression in fourth- and fifth-instar nymphs previously treated with ds*NlInR2* or ds*Nlfoxo*. **(D)** Temporal expression of *Nlvg* in the thorax of wild-type (*wt*) SW and LW BPH strains. **(E)** Tissue distribution of *Nlvg* in fifth-instar nymphs of the *wt* SW-BPH strain. **(F)** *Nlvg* expression in ds*NlInR1*-treated (SW-destined) and ds*gfp*-treated (LW-destined) nymphs. Fifth-instar nymphs (2 hAE) were microinjected with ds*NlInR1*, and then collected at 24, 48, and 72 hAE for quantification of *Nlvg* transcripts via qRT-PCR. Bar represents mean ± s.e.m. derived from three independent biological replicates. Statistical comparisons between two groups were performed using a two-tailed Student's *t*-test (*$P < 0.05$, **$P < 0.01$, and ***$P < 0.001$).

showed that *Nlvg* expression was comparable between the ds*NlInR2*- and ds*gfp*-treated groups at the fourth-instar stage, but was significantly induced in the 48-and 72-hAE fifth-instar nymphs previously treated with ds*NlInR2* (Student's *t*-test, both $P < 0.001$) (Fig 2C). To further confirm this observation, we assessed *Nlvg* expression in the context of *Nlfoxo* knockdown since ds*Nlfoxo* could phenocopy the ds*NlInR2* effect (Fig 1B). The result showed a similar expression of *Nlvg* in ds*Nlfoxo*-treated BPHs to that after *NlInR2* knockdown (Fig 2C). We then investigated whether the *Nl*InR2–*Nl*FoxO signaling cascade affects the expression of additional wing-patterning genes in BPH. To this end, we dissected wing buds from fifth-instar nymphs with *NlInR2* or *Nlfoxo* knockdown and examined the expression level of nine wing-patterning genes including *engrailed* (*en*), *hedgehog* (*hh*), *decapentaplegic* (*dpp*), *apterous* (*ap*), Serrate (*ser*), *wingless* (*wg*), *homothorax* (*hth*), *achaete/scute* (*ac/sc*), and *Distalless* (*dll*). Knockdown of *NlInR2* or *Nlfoxo* had no significant effect (< 2-fold) on the expression level of these genes (S2 Fig and S6 Table), confirming the regulatory specificity of *Nlvg* by the *Nl*InR2-*Nl*FoxO signaling cascade.

Next, we used qRT-PCR to investigate the temporal expression of *Nlvg* in the *wt* SW-BPH and LW-BPH strains, the latter contains > 80% of LW adults. RNA was isolated from the thorax of first- to fifth-instar nymphs and adults, and *Nlvg* expression was quantified via qRT-PCR. Expression levels of *Nlvg* were comparable between SW- and LW-BPH strains through the first four nymphal stages (first to fourth instars), but was dramatically up-regulated in fifth instars (Student's *t*-test, $P < 0.001$) and remained at a high level until the adult stage in LW-BPH strain (Fig 2D). In addition, tissue distribution analysis showed that a relatively high amount of *Nlvg* transcripts were detected in wing buds of fifth instar SW-BPH

(Fig 2E), indicating that *Nlvg* might be needed for the development of both LW and SW morphs. These findings prompted us to investigate whether redirection of wing commitment from LW to SW would lower *Nlvg* expression. To this end, we silenced *NlInR1* in 6-hAE fifth instar *wt* LW-BPH, which should redirect wing development from LW to SW morphs by inactivating the IIS pathway [10]. Compared to ds*gfp*, ds*NlInR1* treatment significantly reduced *Nlvg* expression in 24-, 48-, and 72-hAE fifth instar *wt* LW BPHs (Fig 2F). This evidence suggests that the wing-patterning gene *Nlvg* is most likely a downstream target of the IIS pathway.

## *Nlvg* mediates the ds*Nlfoxo* effect on wing-morph transition

If *Nlvg* is indeed a downstream target of *Nl*InR2–*Nl*FoxO signaling, RNAi-mediated silencing of *Nlvg* (ds*Nlvg*) should reverse LW development in ds*Nlfoxo*-treated nymphs. To test this hypothesis, we first investigated the *Nlvg* knockdown phenotype by microinjecting third-instar *wt* LW BPH nymphs with ds*Nlvg*. Microinjection with ds*Nlvg* significantly reduced the transcriptional level of *Nlvg* compared to ds*gfp* treatment (Fig 3A). The majority of ds*Nlvg*-treated nymphs died before the adult stage (S3 Fig), and the surviving nymphs molted into IMW adults, in contrast to the LW adults that developed after ds*gfp* treatment (Pearson $\chi^2$ test: $\chi^2 = 29.474$, $df = 1$, $P < 0.001$) (Fig 3B and 3C). Statistical analysis showed that the IMW adults had wings conspicuously smaller than those of ds*gfp*-treated LW adults (Fig 3D), although overall body size was similar for the two groups. In addition, silencing of the *vestigial* gene in *Sogatella furcifera*, another planthopper species, resulted in 100% IMW adults ($n = 20$) compared to

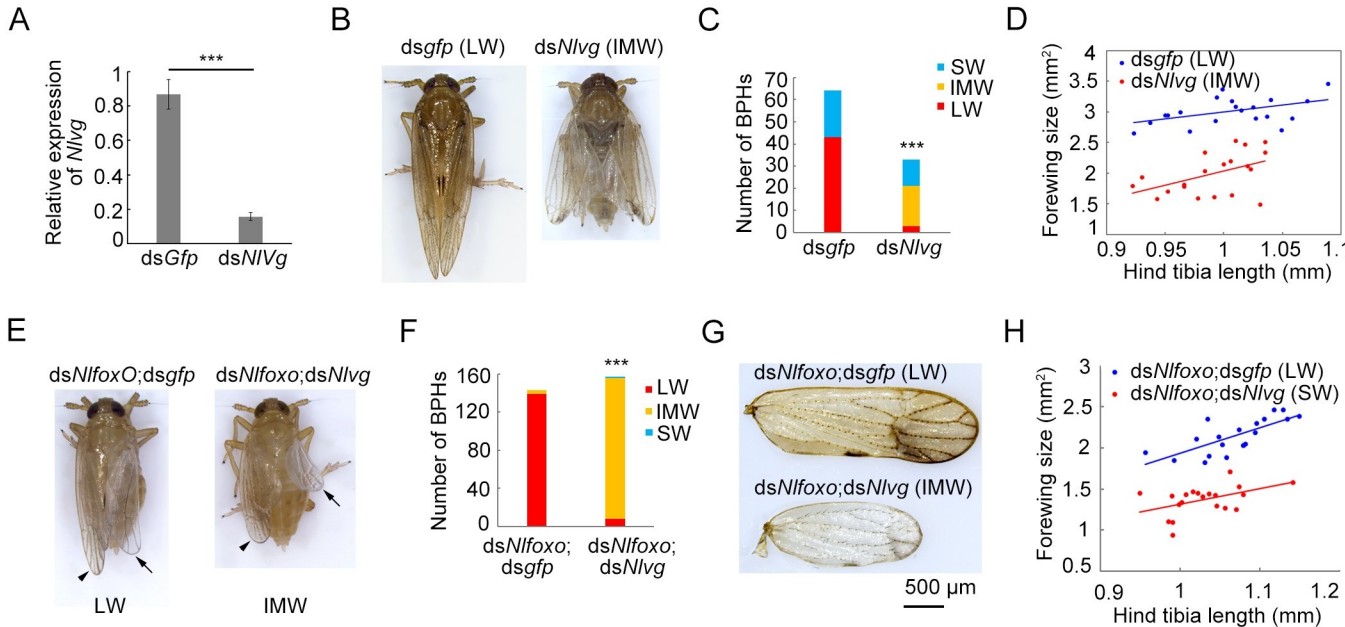

**Fig 3. Knockdown of *Nlvg* reverses the ds*Nlfoxo* effect. (A)** Examination of RNAi efficiency by qRT-PCR. Third-instar nymphs were microinjected with ds*Nlvg* or ds*gfp*, and then fifth-instar nymphs ($n = 5$) were collected for qRT-PCR analysis. The relative expression level of *Nlvg* was normalized to that of the *Nl18s* gene. Bar represents mean ± s.e.m. derived from three independent biological replicates. Statistical comparisons between two groups were performed using a two-tailed Student's *t*-test (***$P < 0.001$). (B) Female adults with *Nlvg* or *gfp* knockdown. LW, long wings. IMW, intermediate-size wings. **(C)** Numbers of BPH adults with different wing morphs treated with ds*Nlvg* or ds*gfp*. ***, $P < 0.001$ (Pearson $\chi^2$ test: $\chi^2 = 29.474$, $df = 1$). **(D)** The relative wing size in BPHs treated with ds*Nlvg* ($n = 20$) or ds*gfp* ($n = 20$). **(E)** Female adults with double-gene knockdown. Third-instar nymphs were microinjected with either ds*Nlfoxo*; ds*Nlvg* or ds*Nlfoxo*;ds*gfp*. Forewings at the right side were removed. Arrow and arrow heads represent hindwings and forewings, respectively. **(F)** Numbers of BPH adults with different wing morphs treated with ds*Nlfoxo*;ds*gfp* or ds*Nlfoxo*;ds*Nlvg*. ***, $P < 0.001$ (Pearson $\chi^2$ test: $\chi^2 = 254.06$, $df = 1$). **(G)** Morphology of forewings in BPHs with double-gene knockdown. **(H)** Relative wing size in BPHs treated with ds*Nlfoxo*;ds*Nlvg* ($n = 20$) or ds*Nlfoxo*;ds*gfp* ($n = 20$). Each dot represents the wing size and tibia length derived from an individual female.

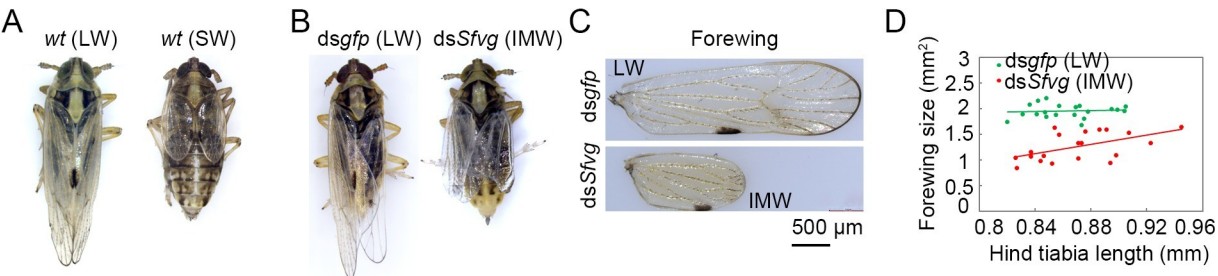

**Fig 4. Knockdown of *S. furcifera vestigial* homologue (*Sfvg*).** **(A)** Wild-type LW and SW *S. furcifera* adults. **(B)** Knockdown of *Sfvg* led to an adult with intermediate-size wings (*n* = 20). **(C)** Morphology of forewings treated with ds*Sfvg* or ds*gfp*. **(D)** Relative wing size in *S. furcifera* treated with ds*Sfvg* (*n* = 20) or ds*gfp* (*n* = 20). Each dot represents the wing size and tibia length derived from an individual female. LW, long-winged. SW, short-winged. IMW, intermediate-size wings.

95% LW adults (*n* = 20) after ds*gfp* treatment (Fig 4). This observation agrees with the phenotype we observed for BPH, underscoring a pivotal and conserved role of the *vestigial* gene in wing development in the planthopper family. Next, we performed a double-gene knockdown assay in third-instar *wt* SW BPH nymphs via microinjection of a dsRNA mixture of ds*Nlfoxo* and ds*Nlvg* (ds*Nlfoxo*;ds*Nlvg*), or ds*Nlfoxo* and ds*gfp* (ds*Nlfoxo*;ds*gfp*). Consistent with our previous report [10], ds*Nlfoxo*;ds*gfp* treatment redirected wing development from SW to LW, leading to LW adults (Fig 3E). By contrast, simultaneous depletion of *Nlvg* and *Nlfoxo* led to IMW adults (Fig 3E), and the wing-morph ratio is significantly different between ds*Nlfoxo*;ds*Nlvg* and ds*Nlfoxo*;ds*gfp* treatments (Pearson $\chi^2$ test: $\chi^2$ = 254.06, *df* = 1, *P* <0.001) (Fig 3F). Adults previously treated with ds*Nlfoxo*;ds*Nlvg* had significantly smaller wings relative to those with ds*Nlfoxo*;ds*gfp* treatment (Fig 3G and 3H). These observations suggest that *Nl*InR–*Nl*FoxO signaling mainly depends on *Nl*Vg to regulate LW development in BPH.

## *Nl*FoxO specifically binds to the first intron region of the *Nlvg* locus

The FoxO transcription factors contain a conserved DNA-binding domain (DBD) of approximately 100 residues in the N-terminal region, and commonly regulate downstream targets mainly by binding to a 13-bp FoxO recognition element (FRE) containing a FoxO consensus binding site (TGTTTAC) [19–25]. The DBD of *Nl*FoxO shares a high degree of sequence similarity with its orthologs in the fruitfly *D. melanogaster*, the nematode *Caenorhabditis elegans*, and even in human *Homo sapiens* (Fig 5A). To investigate whether *Nlvg* represents a direct target gene of *Nl*FoxO, we searched the *Nlvg* locus for FRE regions in SW BPH. This analysis identified five copies of FRE (FRE1–5) distributed in three different introns of the *Nlvg* locus (Fig 5B), but not in the promoter region, even when we examined a 3-kbp segment upstream of the transcription start site of *Nlvg*. Intriguingly, multiple FREs were also identified in additional eight wing-patterning genes (S4 Fig), although the expression levels of these genes were not significantly affected in the context of *NlInR2* or *Nlfoxo* knockdown. We further cloned the FRE1–5 region in *Nlvg*, and confirmed the sequence using Sanger sequencing. The DBD of *Nl*FoxO was then expressed as a maltose-binding protein (MBP) fusion protein (MBP/DBD) in an *Escherichia coli* system, and was purified for an electrophoretic mobility shift assay (EMSA). To easily measure the binding affinity of the MBP/DBD protein, each oligonucleotide (FRE1–5) was labeled with Alexa Fluor 680 at the 5′-end. EMSA showed that MBP/DBD was capable of binding to any one of the FRE1–5 to form protein–DNA complexes in vitro, but the strongest band was observed for FRE1 (Fig 5C). This result indicates specific additional flanking nucleotides of the FoxO consensus binding site are required for efficient binding. To further confirm the binding specificity of MBP/DBD for FRE1, we performed competition

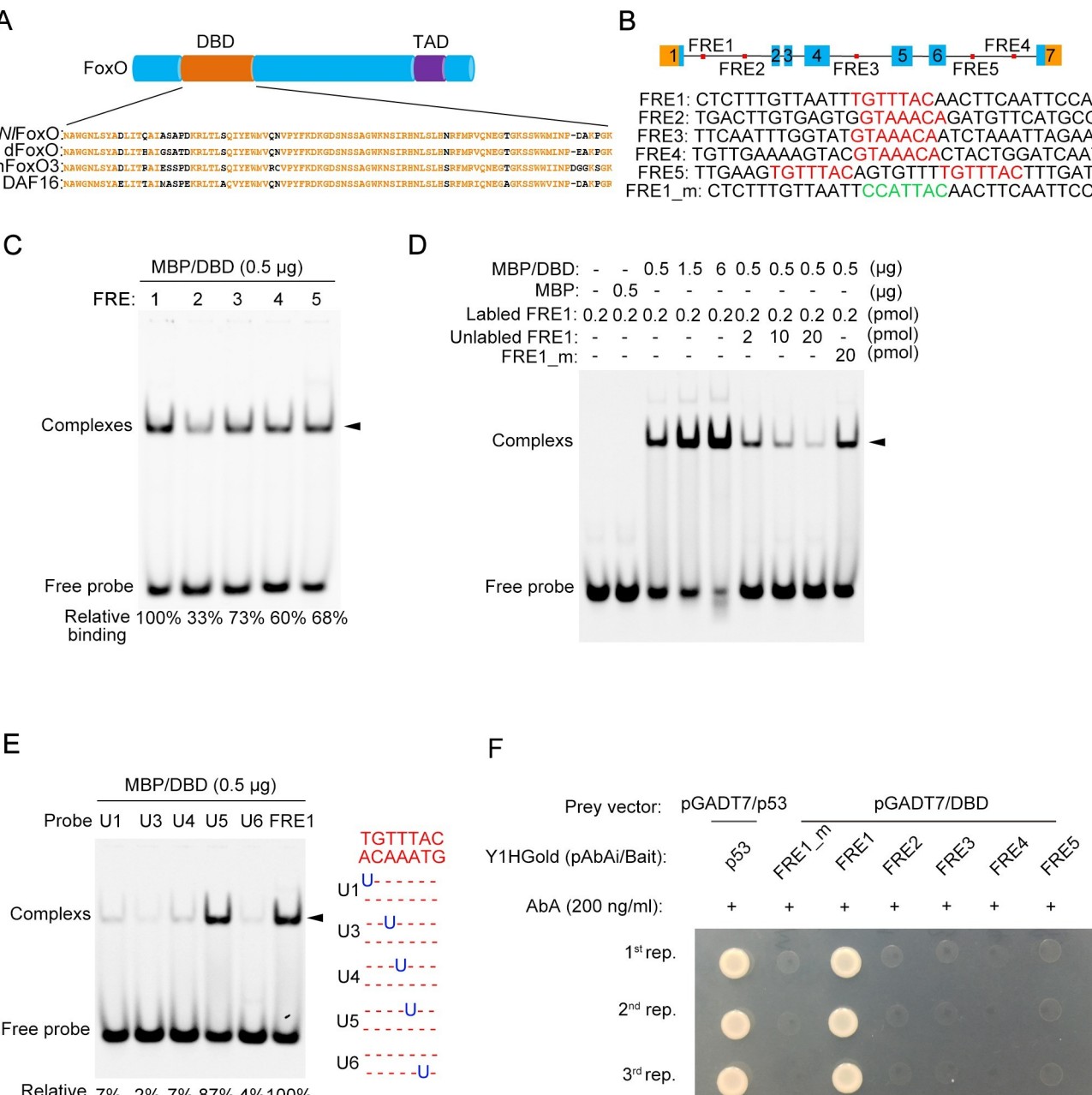

**Fig 5. Electrophoretic mobility shift assay (EMSA).** **(A)** Schematic depiction of the *Nl*FoxO domains and sequence alignment. *Nl*FoxO contains a DNA binding domain (DBD) and a transactivation domain (TAD). The *Nl*FoxO DBD was aligned with its orthologs from *D. melanogaster* (dFoxO), *C. elegans* (DAF16), and *H. sapiens* (hFoxO3). **(B)** Schematic diagram of the distribution of the FoxO recognition element (FRE) in *Nlvg*. The FoxO consensus binding site (TGTTTAC or its complementary sequence, in red) and its flanking sequences are indicated in FRE1–5. The FRE1_m was generated by randomly mutating the FoxO consensus binding site (TGTTTAC) of FRE1 to 'CCATTAC' (in green). **(C)** EMSA of MBP/DBD binding to Alexa Fluor 680-labeled FRE1-5. **(D)** EMSA of the MBP/DBD binding to Alexa fluor 680-labeled FRE1, unlabeled FRE1, and mutant FRE1 (FRE1_m). **(E)** EMAS of the MBP/DBD binding to Alexa fluor 680-labeled FRE1 containing substitutions within the FoxO consensus sequences. The protein-DNA complex is indicated by arrowheads. The relative binding affinity is shown at the bottom. **(F)** Binding of MBP/DBD to FRE1–5 in a yeast one-hybrid assay. Vectors containing p53 and mutant FRE1 (FRE1_m) served as positive and negative controls, respectively. AbA, Aureobasidin A antibiotic. Experiments were performed three times with similar results (1st-, 2nd-, and 3rd-rep.).

binding assays by incubating MBP/DBD with high concentrations of either unlabeled or mutant probes. In the control groups, the probe alone or incubated with the MBP tag protein could not form complexes (Fig 5D). By contrast, a higher density of protein–DNA complex was observed with increasing MBP/DBD protein concentrations (Fig 5D). Incubation of the reaction mixture with increasing concentrations of excess unlabeled competitor probes (10×, 50× and 100×) led to a gradual reduction in complex formation (Fig 5D). However, the protein–DNA complex did not change when a high level of mutated FRE1 probes was added as a competitor (Fig 5D). The FRE1_m was generated by randomly mutating the FoxO consensus binding site (TGTTTAC) of FRE1 (Fig 5B), theoretically losing the ability to compete with FRE1 for FoxO binding. We then investigated whether the FoxO consensus binding site (TGTTTAC) in FRE1 is essential for MBP/DBD binding, as reported previously for human FoxO3. To this end, we substituted individual thymine (T) nucleotides with uracil (U) and measured the binding affinity of MBP/DBD for the mutant probes. The results showed that base substitution at positions U1, U3, U4, and U6 almost abolished formation of the protein–DNA complex, whereas base substitution at position U5 only moderately decreased binding (Fig 5E), thus further confirming the binding specificity of the MBP/DBD.

To further test whether *Nl*FoxO can bind to *Nlvg* FRE1 in vivo, we carried out a yeast one-hybrid assay. Three tandem copies of each FRE (bait) were cloned into the upstream of an *Aureobasidin A resistance* (*AbA<sup>r</sup>*) reporter gene in a pAbAi vector, and then integrated into the genome of the yeast Y1HGold strain to create bait–reporter strains, which were subsequently transformed with the prey vector pGADT7/DBD. We observed that the bait–reporter strains containing FRE1 or p53 (a bait control) could grow on medium containing the antibiotic AbA, whereas neither the FRE1 mutant (FRE1_m) nor FRE2–5 could survive (Fig 5F). Taken together, the in vivo and in vitro binding assays indicate that *Nl*FoxO may suppress *Nlvg* expression by directly binding to FRE1 in the first intron of the *Nlvg* locus.

## Discussion

Although previous studies have shown that the IIS pathway serves as a 'master signal' that determines wing-morph switching via the activity of the transcription factor *Nl*FoxO in BPH, the downstream effectors responsible for IIS activity remains unknown. Our findings reveal that the key wing-patterning gene *Nlvg* is selectively and temporally up-regulated by the IIS pathway during the wing-morph decision stage (fifth-instar stage). Protein–DNA binding assay suggests that *Nl*FoxO protein might suppress *Nlvg* expression via binding to an intronic FoxO response element. By this way, *Nlvg* dysfunction can considerably antagonize *Nl*InR2–*Nl*FoxO signaling activity and redirect wing commitment from LW to IMW morphs.

We used RNA-seq to investigate differentially expressed genes (DEGs) in LW-destined wing buds (ds*InR2* treatment) versus SW-destined wing buds (ds*gfp* treatment) in 24- and 48-hAE fifth instars (Fig 2A). Intriguingly, only one DEG was found between the two groups in 24-hAE fifth instars. Considering that the wing-morph decision period in BPH is in the fifth instar stage (the last nymphal stage), it is plausible that wing buds in LW and SW BPHs might share a common developmental trajectory before the 24-hAE fifth instar. Alternatively, the changes were too small to be detected at the RNA-seq sensitivity available. In 48-hAE fifth instars, differential expression was only observed for a small subset of genes (1.25%, 299 out of 23,916 mapped genes, S1 Data), among which *Nlvg* was significantly up-regulated in LW-destined relative to SW-destined wing buds. In addition, *Nlvg* was significantly induced in the context of *Nlfoxo* knockdown during the fifth-instar stage, indicating that *Nlvg* is a downstream target of the *Nl*InR2–*Nl*FoxO signaling cascade. The function of the *vg* gene has been studied in great depth in *Drosophila*, and its product acts as a trans-activator to control wing

formation and identity [26,27]. Loss of *vg* function in *Drosophila* completely eliminates wing and haltere formation [27], whereas its ectopic expression is sufficient to convert cells in the eye, antenna and leg discs to wing-like fates [15,16], indicating that *vg* plays a pivotal role in wing formation. The *Nl*Vg protein shares high sequence similarity in amino acid residues with the *Drosophila vg* homologue. *Nlvg* was constantly expressed through the developmental stages of BPH, but temporally increased in the fifth-instar stage of LW-BPH strain, but not in the SW-BPH strain. Interestingly, we detected a considerably high level of *Nlvg* transcripts in SW-BPH wing buds relative to other body parts, although the SW adult contains truncated-forewings and rudimentary hindwings, indicating that *Nl*Vg is required for SW development. Moreover, the *Nlvg* expression level could be further increased if wing commitment were changed from SW to LW by ds*NlInR2* or ds*Nlfoxo* treatment; conversely, altering wing commitment from LW to SW by ds*NlInR1* treatment could further decrease *Nlvg* expression. These data indicate that the expressional fluctuation of *Nlvg* is likely regulated by the IIS pathway, and thus might be tightly associated with wing-morph switching.

Most of what we understand about insect wing formation comes from studies in *D. melanogaster*. *Drosophila* wings arise from imaginal discs that are subdivided into anterior–posterior (A-P) and dorsal–ventral (D-V) compartments by the action of the *en* and *ap* selector proteins, respectively. The *dpp* and *wg* morphogen proteins emanate from the A-P and D-V compartment boundaries, respectively, to organize wing growth and patterning via regulation of numerous downstream wing-patterning genes [28–33]. The *vg* gene is a 'nodal point' that connects compartmentalization to the control of wing formation and identity [16]. Different from the holometabolous insect *D. melanogaster*, wing development in BPH is a gradual process, in which wing buds enlarge at each subsequent molt. Here, we found that RNAi-mediated silencing of *Nlvg* partially but not completely reversed the ds*Nlfoxo* effect, resulting in IMW morphs rather than SW morphs. We speculate that IMW might be derived halted development of wing buds, which had started growth at earlier nymph stages. In addition, we examined the expression level of an additional nine principal components of the wing-patterning network in the context of *NlInR2* or *Nlfoxo* knockdown. Consist with the RNA-seq data showing that only *Nlvg* was selectively regulated, no significant change (< 2 fold) was observed for these genes, although *NlInR2* or *Nlfoxo* knockdown indeed led to wing commitment from SW to LW. This phenomenon is reminiscent of the regulation of wing morphs by wing-patterning genes in ants, where winged and wingless ants likely evolved from interruption points in the wing gene network [34,35]. The expression of several genes within the wing-patterning network differed among different ant species and even different castes in the same species. Pointes of interruption within the wing-patterning network might halt wing development, leading to wingless worker castes [34].

FoxO transcription factors usually activate or suppress gene expression via direct binding to DNA target sites and interaction with other effectors [36,37]. Both in vitro and in vivo assays strongly indicate that *Nl*FoxO showed strongest binding capacity to FRE1 located in the first intron region of the *Nlvg* locus. Based on this finding, we draw an appealing conclusion that the IIS pathway regulates *vg* expression through FRE1. A potential limitation to this conclusion is that we are unable to generate a FRE1-null BPH mutant because of a high ratio of single nucleotide polymorphisms in intron 1 of the *Nlvg* locus. This intronic binding by FoxO has been reported in several other animal model systems such as *Drosophila*, mouse, and human tumor cells [38–40]. Coincidently, *Drosophila vg* expression is synergistically modulated by the D-V and A-P axes in wing discs via sequential activation of the second and fourth intronic regulatory enhancers in the *vg* locus [16,18,41–43], respectively. Whether intronic regulation of *vg* expression is a common mechanism adopted by transcription regulators across diverse taxa remains to be addressed.

In conclusion, our results highlight that the key wing-patterning gene *Nlvg* is a main downstream effector that mediates the IIS activity on LW development during the wing-morph decision stage. To the best of our knowledge, our findings provide a first glimpse of the link connecting the IIS pathway to the wing-patterning network in regulating wing polymorphism in insects.

## Materials and methods

### Insects

The SW-BPH strain (SW ratio > 90%) was originally collected from a rice field in Hangzhou, China, in 2008. The LW-BPH strain (LW ratio > 80%) was kindly provided by Dr. Hongxia Hua (Huazhong Agriculture University, Wuhan, China). The SW-BPH strain was purified by sibling inbreeding for 13 generations, and was previously used for genomic DNA sequencing and assembly [44]. Insects were reared in a walk-in chamber at 26 ± 0.5°C under a photoperiod of 16 h light/8 h dark at relative humidity of 50 ± 5% on rice seedlings (rice variety Xiushui 134).

### RNAi and microinjection

dsRNA synthesis and injections were performed as previously reported [10,45]. In brief, dsRNA was synthesized using a T7 RNAi transcription kit (Vazyme) according to the manufacturer's instructions. The dsRNA primers for the corresponding genes were synthesized with a T7 RNA polymerase promoter at both ends (S1 Table). Microinjection was performed using a FemtoJet microinjection system (Eppendorf). Insects were anesthetized with carbon dioxide for 10–15s before microinjection. After injection, all insects were maintained in transparent polycarbonate jars with fresh rice seedlings. At 2 days after injection, insects (*n* = 5 for each of three replicates) were collected for RNA extraction, and cDNA synthesis for assessment of RNAi efficiency via qRT-PCR.

### qRT-PCR

Total RNAs were isolated from samples using RNAiso Plus (Takara). First-strand cDNA was synthesized from total RNAs (450 ng) using HiScript QRT SuperMix (Vazyme). The synthesized cDNAs were diluted ten-fold and used as templates for qRT-PCR with primers specific for the genes being investigated (S1 Table). qRT-PCR was conducted on a CFX96 real-time PCR detection system (Bio-Rad) with the following conditions: denaturation for 3 min at 95°C, followed by 40 cycles at 95°C for 10s, and then 60°C for 30s. The ribosomal 18S rRNA gene (*Nl18s*) was used as the internal reference gene. The $2^{-\Delta\Delta Ct}$ method (Ct represents the cycle threshold) was used to measure relative expression levels [46]. Three biological replicates were used for statistical comparison between samples.

### Rapid amplification of cDNA ends (RACE) and sequence analysis

Total RNA was extracted from the thorax of LW-BPH individuals (*n* = 15) using RNAiso Plus (Takara). For 5′ RACE, 500 ng of total RNAs was converted into 5′ RACE-ready first-strand cDNA by using SMARTScribe reverse transcriptase and SMARTer II A oligonucleotide (Takara). Then, the 5′-end sequence of *Nlvg* was amplified using primer Nlvg-5RACE-gsp1 (S1 Table) and ligated to the pRACE vector for Sanger sequencing. For 3′ RACE, first-strand cDNA was synthesized from 500 ng of total RNA with the 3′ RACE adaptor primer. Then, nested PCR was carried out to obtain the 3′-end sequence of *Nlvg* with primer pairs consisting of the 3′ RACE outer primer/Nlvg-3RACE-gsp1 and 3′ RACE inner primer/Nlvg-3RACE-np1

(S1 Table). The amplified sequences were cloned and confirmed via Sanger sequencing. The intron and exon regions were predicted by searching for the *Nlvg* cDNA sequence against the BPH genome using the Splign algorithm (https://www.ncbi.nlm.nih.gov/sutils/splign/splign.cgi).

The domains of the deduced amino acid sequence of *Nl*FoxO were predicted using SMART (http://smart.embl-heidelberg.de/). The amino acid sequences of FoxO homologs from *D. melanogaster*, *C. elegans*, and *H. sapiens* were downloaded from GenBank and aligned with *Nl*FoxO using ClustalX (v. 2.1).

## Determination of the wing-morph decision period

SW-destined nymphs were collected at 6-h intervals after fifth-instar ecdysis, and used for dsRNA microinjection. Each fifth-instar individual was microinjected with approximately 150 ng of ds*NlInR2* or ds*gfp*. Two days after microinjection, RNAi efficiency was examined via qRT-PCR from three biological replicates comprising five nymphs each. The remaining nymphs were allowed to emerge into adults, and the number of adults with each wing morph was counted.

## Spatio-temporal expression of *Nlvg* in *wt* BPHs

To investigate temporal expression of *Nlvg*, total RNA was isolated from the thorax of first-instar ($n = 100$), second-instar ($n = 50$), third-instar ($n = 50$), fourth-instar ($n = 30$), fifth-instar nymphs ($n = 15$), and adult females ($n = 15$) at 48 hAE. To examine the tissue distribution of *Nlvg*, antenna, head, fat body, gut, wing buds, leg, and cuticle were dissected from fifth-instar nymphs ($n = 50$, 48 hAE) of SW-BPHs and used for RNA extraction. Three independent biological replicates were used for RNA isolation, and first-strand cDNA was synthesized for quantification of *Nlvg* expression. The relative expression level of *Nlvg* was normalized to that of the *Nl18s* gene.

## Determination of *Nlvg* expression under regulation of the IIS pathway

Third-instar SW-destined nymphs were microinjected with approximately 75 ng of ds*NlInR2* or ds*Nlfoxo*. It is worth noting that knockdown of *NlInR2* or *Nlfoxo* can switch wing commitment from SW to LW. Total RNA was isolated when nymphs ($n = 15$ for each biological replicate) reached the 48-hAE fourth-instar, and 24-, 48-, and 72-hAE fifth-instar stages. Then, first-strand cDNA was synthesized and *Nlvg* expression was compared to that in the control groups (ds*gfp* treatment) via qRT-PCR.

Fifth-instar LW-destined nymphs (2 hAE) were microinjected with ds*NlInR1* to investigate switching of wing commitment from LW to SW. At 24-, 48-, and 72hAE, nymphs ($n = 15$ for each biological replicate) were collected for RNA isolation, and then cDNA was synthesized for quantification of *Nlvg* transcripts via qRT-PCR. To confirm the dsRNA effect, the remaining nymphs were allowed to molt into adults for morphological examination.

## Double-gene knockdown

To investigate whether wing fate could be changed from LW to SW during the fifth-instar stage, fifth-instar nymphs at 2 hAE were microinjected with ds*NlInR2* (150 ng). Then ds*NlInR2*-treated nymphs were further microinjected with ds*NlInR1* (150 ng) or ds*gfp* (150 ng) at 24- or 48-hAE. Nymphs were allowed to emerge into adults, and the number of adults with each wing morph was counted.

To investigate whether *Nlvg* knockdown could reverse the effect of ds*Nlfoxo* on LW development, third-instar nymphs were microinjected with a dsRNA mixture of ds*Nlfoxo*;ds*Nlvg* and ds*Nlfoxo*;ds*gfp*. Each third-instar individual was microinjected with approximately 90 ng of the dsRNA mixture (45 ng for each dsRNA). Nymphs were allowed to emerge into adults, and the number of adults with each wing morph was counted.

## Sample preparation for RNA-seq

To redirect wing development from SW to LW morphs, fifth-instar SW-destined nymphs (2 hAE) were microinjected with 150 ng of ds*NlInR2*. Because wing buds could not be completely separated from the notum, we dissected tissue including the mesonotum, metanotum, and wing buds for RNA isolation at 24 hAE (denoted as dsInR2_24h and dsgfp_24h) and 48 hAE (denoted as dsInR2_48h and dsgfp_48h) from three independent biological replicates ($n = 30$ for each biological replicate). The remaining nymphs were allowed to molt into adults for morphological examination.

## RNA isolation, cDNA library preparation and Illumina sequencing

Total RNA was isolated using RNAiso Plus (Takara) according to the manufacturer's protocol. The RNA quality was examined via 1% agarose gels electrophoresis and a NanoPhotometer spectrophotometer (IMPLEN). RNA integrity was assessed using the RNA Nano 6000 Assay Kit for the Agilent Bioanalyzer 2100 system (Agilent Technologies).

A total of 1.5 μg of RNA per sample was used to construct the sequencing library using a NEBNext Ultra RNA Library Prep Kit for Illumina (NEB) according to manufacturer's recommendations, and index codes were added to each sequence. Library fragments of 250–300bp in length were preferentially purified using an AMPure XP system (Beckman Coulter), and library quality was assessed using the Agilent Bioanalyzer 2100 system. Clustering of the index-coded samples was performed on a cBot Cluster Generation System using a TruSeq PE Cluster Kit v3-cBot-HS (Illumina) according to the manufacturer's instructions. The cDNA libraries were sequenced on an Illumina Hiseq platform and 150-bp paired-end reads were generated.

## Read mapping and DEGs

After Illumina sequencing, clean reads were generated by removing adapters, ploy-N, and low quality reads from the raw data. BPH genome data (GCA_000757685.1_NilLug1.0) were used as the reference sequence for the transcriptomic analysis. Paired-end clean reads were mapped to the reference genome using Hisat2 [47]. The read numbers mapped to each gene were counted using HTSeq (v0.9.1) [48], and gene expression was calculated using the number of fragments per kilobase of transcript sequence per millions base pairs sequenced (FPKM) [49]. The mapped reads for each sample were assembled using cufflinks in a reference-based approach, and then novel genes were predicted by cufflinks. The edgR package was used for differential expression analysis of dsInR2_24h versus dsgfp_24h, with an adjusted *P*-value $< 0.05$ and a fold change $> 2$ set as the thresholds for significant difference in expression. Differential expression analysis of dsInR2_48h versus dsgfp_48h (three biological replicates per condition) was performed using the DESeq R package and gene expression changes were considered to be significantly different when the adjusted *P*-value was $< 0.05$.

To validate the RNA-seq results, 12 DEGs were randomly selected, and their expression levels were measured via qRT-PCR using specific primers for each gene (S1 Table).

## GO and KEGG enrichment analysis of DEGs

Gene Ontology (GO) enrichment analysis of DEGs was carried out using the GOseq R package [50]. KOBAS software [51] was used to assess the statistical enrichment of DEGs in KEGG pathways.

## Expression and purification of MBP/DBD fusion protein

The encoding sequence of the *Nl*FoxO DBD was amplified using DBD-F/DBD-R primers, and then cloned into the pMAL-c5x expression vector (New England Biolabs) with a maltose-binding protein (MBP) at the N-terminus. The recombinant plasmid was transformed into *E. coli* strain Rosetta (DE3). The MBP/DBD fusion protein was expressed in *E. coli* using 0.3 mM IPTG at 37˚C for 2 h for induction. Proteins were purified from cell lysates using the pMAL protein fusion and purification system (New England Biolabs) according to the manufacturer's instruction. Purified proteins were quantified using a Pierce BCA protein assay kit (Thermo Scientific). In parallel, MBP protein was expressed from the pMAL-c5x vector alone and then purified under the same condition.

## Electrophoretic mobility shift assay

An EMSA assay was performed essentially as previously reported [52]. In brief, to prepare labeled double-stranded DNA probes, oligonucleotides (FRE1–5) were labeled at the 5′-end with Alexa Fluor 680. Equal amounts of labeled oligonucleotides and antisense oligonucleotides were mixed to a final concentration of 50 μM in annealing buffer (1 mM EDTA, 0.1 M NaCl, 10 mM Tris–HCl, pH 7.5) and then heated to 94˚C for 10 min, followed by cooling at room temperature for 1.5 h to produce Alexa Fluor 680-labeled probes. EMSA experiments were performed using an EMSA/Gel-Shift Kit (Beyotime) according to the manufacturer's instructions. In brief, 0.2 pmol of labeled probes were incubated with 0.5, 1.5, or 6 μg of the purified MBP/DBD recombinant protein in 10 μl of binding reaction mixture. For the control group, labeled probes (0.2 pmol) were mixed with MBP tag proteins (0.5 μg) only in 10 μl of binding reaction buffer. For competition assays, the purified MBP/DBD recombinant protein (0.5 μg) was incubated with 10-fold (2 pmol), 50-fold (10 pmol) or 100-fold (20 pmol) unlabeled competitor probes before adding the labeled probes, and then placed at room temperature for 20min. All assays were incubated at 25˚C for 20 min then electrophoresed in 6% polyacrylamide gels in 0.5 × TBE for 1h. Gels were imaged using the ChemiDoc MP imaging system (BioRad). The sequences of labeled and unlabeled oligonucleotides are listed in S1 Table.

## Yeast one-hybrid assay

Yeast one-hybrid assay were performed using the Matchmaker Gold Yeast One-Hybrid Library Screening system (Clontech) according to the manufacturer's manual. To generate the bait construct, three tandem repeated sequences of FRE1, FRE2, FRE3, FRE4, FRE5, or FRE1 mutant (FRE1_m) were inserted individually upstream of the *Aureobasidin A resistance* (*AbA^r*) reporter gene in a pAbAi vector that was previously linearized with *Kpn* I and *Xho* I restriction endonucleases. Bait constructs were then integrated into the genome of the Y1HGold yeast strain to generate bait-specific reporter strains Y1HGold (pAbAi/Bait). Each bait-specific reporter Y1HGold strain was suspended in 0.9% NaCl, and adjusted to an optical density of ~0.002 at 600 nm. Then 100-μl aliquots of the suspension were plated on SD/−Ura medium at AbA concentrations of 0, 100, 150 and 200 ng/ml, and then incubated at 30˚C for 2–3 days to confirm the minimal inhibitory concentration of AbA for bait-reporter yeast. The

encoding region of the *Nl*FoxO DBD domain was amplified using DBD-prey-F/R primers (S1 Table), and then inserted downstream of GAL4 in pGADT7 to create a prey vector (pGADT7/ DBD). The prey vector was transformed into bait-specific reporter strains, and then yeast transformants were dropped onto SD/−Leu media plates with or without AbA antibiotic (200 ng/ml). A prey vector and a bait-specific reporter strain containing the p53 gene were used as a positive control.

### Image acquisition and processing

Images of insects were taken using a DVM6 digital microscope (Leica) with LAS X software. Images of wings, and tibia were captured with a DFC320 digital camera attached to a Leica S8AP0 stereomicroscope using the LAS (v. 3.8) digital imaging system. Digital images of fore-wings (*n* = 20) and hind tibias (*n* = 20) were collected for measurement of forewing size and hind tibia length using ImageJ (v. 1.47).

### Statistics

Statistical analysis was performed using SPSS (v. 22) or GraphPad Prism (v. 5.01). Pearson Chi-Square test was used for statistical analysis for Figs 1D, 1E, 3C and 3F and S1 Fig. Two-tailed Student's *t*-tests were used for statistical analysis for Fig 2C, 2D and 2F and S5 Fig. Data are presented as mean ± standard error of the mean (mean ± s.e.m) for three independent bio-logical replicates. A log-rank (Mantel-Cox) test was used for statistical analysis for S3 Fig. Sig-nificance levels are indicated as $P < 0.05$ (*), $P < 0.01$ (**), or $P < 0.001$ (***).

### Supporting information

**S1 Fig. Numbers of male adults with different wing morphs after ds*NlInR2* treatment.** Fifth-instar male nymphs collected at designated time (6, 12, 18, 24, 30, 36, 42, and 48 hAE) were microinjected with ds*NlInR2* or ds*gfp*. hAE, hours after ecdysis. SW, short-winged. IMW, intermediate-size wings. LW, long-winged. Non-significant (n.s.) and significant (*$P < 0.05$, ***$P < 0.01$, ***$P < 0.001$, Pearson's $\chi^2$ test) differences from the control group (ds*gfp*) are indicated.
(TIF)

**S2 Fig. Expression of wing-patterning genes in the context of *NlInR2* or *Nlfoxo* knockdown during the fifth-instar stage.** Third-instar SW-destined nymphs were microinjected with approximately 75 ng of ds*NlInR2* or ds*Nlfoxo*. Total RNA was isolated when nymphs (*n* = 15 for each biological replicate) reached the 24, 48, and 72 hAE fifth-instar. First-strand cDNA was synthesized, and the expression of wing-patterning genes was compared to that in the con-trol groups (ds*gfp* treatment) via qRT-PCR. Bars represent mean ± s.e.m. derived from three independent biological replicates. Statistical comparisons between two groups were performed using a two-tailed Student's *t*-test (*$P < 0.05$, **$P < 0.01$, and **$P < 0.01$).
(TIF)

**S3 Fig. Survival rate of nymphs with *Nlvg* knockdown.** Third-instar nymphs were injected with ds*Nlvg* (*n* = 60) or ds*gfp* (*n* = 60), and surviving BPHs were monitored every 12 h. ds*Nlvg* treatment led to higher mortality relative to the ds*gfp* treatment (log-rank Mantel-Cox test, $P < 0.001$).
(TIF)

**S4 Fig. Schematic depiction FRE distribution in nine wing-patterning genes.** Exons were indicated by boxes in blue. The number of FoxO recognition element (FRE) containing a

FoxO consensus binding site (TGTTTAC) was labeled in red number.
(TIF)

**S5 Fig. Confirmation of RNA-seq by qRT-PCR.** (**A**) Total RNA was isolated from fifth instar nymphs at 48 hAE previously treated with ds*NlInR2* or ds*gfp*. First-strand cDNA was synthesized, and the expression of each gene was compared to that in the control groups (ds*gfp* treatment) via qRT-PCR. (**B**) The number of fragments per kilobase of transcript sequence per millions base pairs sequenced (FPKM) showed by RNA-seq. Bars represent mean ± s.e.m. derived from three independent biological replicates. Statistical comparisons between two groups were performed using a two-tailed Student's *t*-test (*$P < 0.05$, **$P < 0.01$, and ***$P < 0.001$).
(TIF)

**S1 Table. Oligonucleotides used in this study.**
(XLSX)

**S2 Table. Data quality of RNA-seq.**
(XLSX)

**S3 Table. DEGs between dsInR2_48h and dsgfp_48h.**
(XLSX)

**S4 Table. GO classification of differentially expressed genes.**
(XLSX)

**S5 Table. KEGG enrichment of differentially expressed genes.**
(XLSX)

**S6 Table. The expression level of wing-patterning genes.**
(XLSX)

**S1 Data. Transcriptomic analysis of wing buds.**
(DOCX)

## Acknowledgments

We thank Miss Dan-Ting Li for preparing Fig 2A.

## Author Contributions

**Conceptualization:** Hai-Jun Xu.

**Data curation:** Jin-Li Zhang, Sheng-Jie Fu, Sun-Jie Chen.

**Formal analysis:** Jin-Li Zhang, Hao-Hao Chen, Hai-Jun Xu.

**Funding acquisition:** Hai-Jun Xu.

**Investigation:** Jin-Li Zhang, Sun-Jie Chen, Yi-Lai Liu, Xin-Yang Liu.

**Project administration:** Hai-Jun Xu.

**Resources:** Hai-Jun Xu.

**Supervision:** Hai-Jun Xu.

**Visualization:** Jin-Li Zhang.

**Writing – original draft:** Jin-Li Zhang, Hai-Jun Xu.

**Writing – review & editing:** Jin-Li Zhang, Hai-Jun Xu.

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
