## [Decision Letter · Decision Letter 0]

12 Nov 2020

Dear Dr Xu,

Thank you very much for submitting your Research Article entitled 'Vestigial mediates the effect of insulin signaling pathway on wing-morph switching in planthoppers' to PLOS Genetics. Your manuscript was fully evaluated at the editorial level and by independent peer reviewers. The reviewers appreciated the attention to an important topic but identified some aspects of the manuscript that should be improved.  I believe the paper can be published without additional experiments, so I will leave it up to you whether to identify direct FREs at other loci (if that would require new experiments).  However, please do add the quantification of the vg level following the RNAi knock-down, and add the sequence and strain information for the FRE experiments as requested by Reviewer 3.  I also recommend that you add a little bit more information about planthopper insulin receptors to the Introduction, and I agree that the Discussion section would benefit from a brief discussion of wing polyphenism in ants.

We therefore ask you to modify the manuscript according to the review recommendations before we can consider your manuscript for acceptance. Your revisions should address the specific points made by each reviewer.

[LINK]

Yours sincerely,

Artyom Kopp

Associate Editor

PLOS Genetics

Bret Payseur

Section Editor: Evolution

PLOS Genetics

Reviewer's Responses to Questions

**Comments to the Authors:**

Reviewer #1: The manuscript by Zhang et outlines a really exciting study about the genetic regulation of wing polyphenisms in brown plant hoppers. This work follows on from very high profile work showing that in these insect there are two types of insulin receptors, a canonical InR (InR1) and a second InR (InR2) that when it heterodimerises with InR1 it inhibits insulin signalling and directs wing growth toward the short wing morph. This work provides evidence that InR2 is able to exert these effects via interactions between Foxo and Vestigal, a gene central to the wing patterning network. The authors derived their conclusions through detailed studies of vestigial expression, both in the wild type strains and when InR2, Foxo, and vestigial are knocked down with RNAi. Further, they provide biochemical and in vivo evidence (from yeast) that Foxo binds directly to the vestigial regulatory regions. This is a very thorough and interesting study that I think makes significant advances to our understanding how polyphenisms are regulated at the level of gene networks. I have some comments that I feel will improve the readability of the manuscript.

For readers less familiar with the brown plant hopper insulin receptors, I think a bit more detail in the introduction about how InR1 and InR2 work would be helpful. My understanding is that InR1 signalling is akin to insulin signalling in other insects (and indeed other animals). InR2 is unique in that it represses insulin signalling in the wing discs, causing the development of short wing morphs. This is really counterintuitive for those of us who don’t work on plant hoppers. A diagram of the genetic network would really help.

Lines 125-126: 6-24 hAE interval (and 24 to 48 h AE interval on line 128) refers to the 5th instar right? The previous two sentences refer to injections in 3rd and 4th instar nymphs, so it might be worth making this clear.

In Lines 167-168: It would be useful to provide the reader with some indication of why you chose to knock down foxo, and not some other component of insulin signalling. As it stands the connection between InR2 and foxo is not clear. This again relates to the comment above, I think a more thorough discussion of InR1, InR2, and insulin signalling in the plant hopper wing would really help.

There has also been some beautiful work on the genetic networks regulating wing polyphenism in other insects, notably ants (see Abouheif and Wray 2002. Oettler et al 2019). I think it would be useful to contrast what is known about the genetic regulation of wing polyphenisms in ants to that of the brown planthopper.

Minor comments:

Line 76: Change to “… OBLIGATELY flightless.”

Line 124-125: Change to “… generated AS high a proportion”

Line 143: Change to “…, wing buds ARE DESTINED…”

Line 315: Change to “… resulting IN IMW…”

Line 317: Change to “… of AN additional nine principal…”

Line 328: Change to “… conclusion that the IIS pathway REGULATES Vg expression…”

Reviewer #2: This is an important contribution to the literature on phenotypic plasticity in animals and the genetic mechanisms that link environmental variation to phenotypic variation. These authors present a significant contribution to our understanding of how the insulin signaling pathway interacts with the downstream effector gene Vestigial that results in a phenotype appropriate to the environment.

Specifically, these authors have put together a clear line of evidence that the insulin signaling pathway, shown by these authors and others, transduces a nutritional signal in brown rice planthopper post-embryonic development that regulates the expression of the wing patterning gene Vestigial. They found this gene with a comparative transcriptomic screen using an InR2 knockdown and a GFP knockdown control and a time series to identify critical periods of development. The InR2 knockdown results in a long wing phenotype and the GFP knockdown results in a short wing phenotype. Comparison of gene expression at two different time points post injection allowed them to identify Vg as a potentially important downstream gene in this cascade.

The experiments they describe to confirm and further validate their initial hypothesis that Vg is in fact a downstream target are well done and carefully described. In fact, they also checked to see if other wing patterning genes were impacted by InR2 knockdown or Foxo knockdown but of the 9 genes they checked, Vg was the only one that was directly impacted by InR2 or Foxo manipulation. This is a very important part of this story. Wing patterning genes that are linked to insulin signaling directly is what makes this particular study so exciting.

In fact, its the EMSA in vivo and in vitro binding assays that make this story really exciting. Showing that NlVg has FoxO recognition elements that bind to the DNA Binding domain of nlFoxO and then testing this using the binding assays is really great. Very few evo-devo studies in non-model organisms have shown this level of support for interactions directly.

It's interesting to note the differences between Vg function in Drosophila (a holometabolous insect whose imaginal discs develop synchronously) to Vg function in the planthopper that is a hemimetabolous insect whose imaginal wing buds develop over time. The discussion is well written and the discussion of Vg and its role in the planthopper is solid. However, I think the authors should consider in the paragraph starting with line 306 that the manipulation of Vg expression during wing bud development in planthoppers will NOT be like Drosophila. Short wing planthoppers have wings. The nymphs have wing buds. They are just truncated or arrested in their development. Therefore, developmentally, it makes sense that Vg has a role and is expressed and that manipulating Vg at the 5th instar results in intermediate wings. Wing development has already started! Truncation of proliferation or of development occurs once the manipulation occurs but whatever occurred up until that point is still present.

Specific comments:

The title of this paper is confusing. I recommend rewriting it to make it more clear. Vestigial does mediate the effect of insulin signaling on wing development but it might be better to say this more directly. Something like "Insulin signaling regulates wing polyphenism by directly regulating the wing patterning gene Vestigial in long and short wing planthoppers."

Line 197 and also Figure 3 - the use of the word "neutralize" seems strange to me. I think you mean that knockdown of Vg reverses the effect of FoxO and results in stunted or arrested wing development. What your results show is that Vg begins to act, and wings begin to develop but that the loss of Vg through RNAi stops wing cells from proliferating resulting in intermediate wing phenotypes.

The inclusion of the data from Sogatella furcifera is also very interesting. Of course, the inclusion of this data does not also show that the insulin signaling pathway is linked to Vg in S. furcifera but the implication is that it would be. Without the strong links to the insulin signaling pathway from N. lugens, the RNAi data for Vg knockdown would seem to be further evidence that Vg is involved in wing development. The interesting part of this story is that the insulin signaling pathway directly regulates Vg expression.

Line 315 - the use of the word neutralize is again not really correct in this context.

Reviewer #3: In this study, Zhang et al. found that vestigial (vg), a gene encodes a transcription cofactor that plays critical roles in the Drosophila wing development, is a direct target of the insulin/insulin-like growth factor signaling (IIS) pathway in the context of phenotypic plasticity of wing size in planthoppers. Environmentally- and sexually-controlled phenotypic plasticity is widely known in insects and other animals, but the underlying genetic mechanisms are poorly understood. This work is based on the previous pioneering finding, that two insulin receptors play central roles to form short- and long-winged planthoppers. Taking advantage of this unique developmental model, this work makes significant progress on the previous study by showing that particular cis-regulatory elements of a core wing patterning gene vg are direct targets of the evolutionarily conserved IIS effector FOXO. The data in this study will facilitate further understanding of genetic and evolutionary mechanisms underlying phenotypic plasticity in insects and other animals. For these reasons I recommend the publication of this study as an article in PLoS Genetics, but before the publication I suggest authors add some more data to support the claim that the IIS specifically targets vg in this developmental context, and to further address the extent of contribution of the vg locus on the wing dimorphism in this species. Each point that might be addressed is listed below:

Major points

Line 220

It is an important point of this study whether the direct regulation by FOXO is specific to the vg locus or not. Authors address this point by checking the expression level of other known wing patterning genes in nota and conclude that IIS specifically regulates vg in the context of the wing phenotypic plasticity. To further address this point, it would be informative to show distributions of FREs in a couple of other gene loci that authors examined expression level. Please add this data as supplementary material, and discuss differences in the number of FRE between vg and other wing patterning genes.

Line 228

Please describe the sequence source from which these FREs are identified. It is important to clarify whether it is from LW or SW strain. Please add the sequence data from the other strain, and discuss whether the differences in the FREs at the vg locus is responsible for the wing dimorphism in this species.

Furthermore, if possible, please show FRE distributions in vg loci of a few other phylogenetic relatives that have no plasticity in wing size, and discuss whether the number or sequence of FREs is critical for the plasticity.

These data may support the authors’ claim that vg is the central mediator of IIS signal in the context of wing phenotypic plasticity.

Line 313

Please add the data that show the vg expression level after dsRNA injection. Generally, the effect of RNAi treatment depends on a target gene and dsRNA sequence. The intermediate wing phenotype after vg dsRNA injection could be due to the weak effect of the RNAi treatment instead of the involvement of other genes.

Line 328

Does a “high ratio of single nucleotide polymorphisms” mean SNPs in FRE sequences? If it is the case, a comparison of the first intron sequences from both SW and LW strains would be added in the result section, and discuss the possible contribution of this genomic region to the wing dimorphism.

Minor points

Lines 31, 171

Capital or small letters are appropriately used as the first character of the gene name. It indicates if a gene is dominant or recessive. In general, an ortholog of a Drosophila gene in other insect species follows how it is spelled in the Drosophila gene.

e.g. Vestigial  vestigial; Engrailed  engrailed

Line 171, S2 Fig.

Were the same tissues as used for RNA-seq analysis (i.e. T2 and T3 nota and wing pads) sampled for this qPCR analysis? Please clearly describe tissues used for this qPCR analysis.

In case different tissues were used for these analyses, the expression level of the nine patterning genes could be shown with normalized expression values (e.g. TPM, FPKM) from the RNA-seq data to support the authors’ claim.

Line 111

It would be helpful to describe how long the fifth instar lasts under the normal rearing condition for showing that the IIS sensitive period is at the early stage of the fifth instar.

Line 136, Figure 1D

A “double-gene RNAi assay (dsNlInR2;dsNlInR1)” sounds both dsRNAs were injected at the same time. It might be better to briefly explain here that dsInR2 and dsInR1 are injected sequentially in different timing as the authors describe in the legend of Figure 1. It is also helpful to insert a schematic that indicates the time course of this experiment in Figure 1D.

Line 175

SS Fig  S2 Fig

Line 247

Please describe how the mutated FRE1 probe is synthesized. Is it point-mutated or randomly mutated? It is unclear why the mutated probe could not prevent a DNA-protein binding in the EMSA. This point should be explained.

Line 325

Please revise this statement because it sounds contradictory to the result of the in vitro assay which indicates the binding of FOXO to all FREs tested.

Figs. 1, 3, 4: Images of planthoppers

Please consider adding labels that indicate the strain of planthoppers (either SW or LW strain) shown in each panel. It would help readers to understand the data.

S2 Fig.

Please revise the legend. There is no data of vg expression here.

S4 Fig.

What is the number on the y-axis of this plot? How are values from qPCR and FPKM compared in the same axis? Please clarify these points and split plots if needed. Besides, please label each plot with an annotated gene name with a transcript ID. Please mention this figure at an appropriate position in the manuscript.

**Have all data underlying the figures and results presented in the manuscript been provided?**

Reviewer #1: Yes

Reviewer #2: Yes

Reviewer #3: Yes

PLOS authors have the option to publish the peer review history of their article (what does this mean?). If published, this will include your full peer review and any attached files.

Reviewer #1: No

Reviewer #2: No

Reviewer #3: **Yes: **Takahiro Ohde

---

## [Decision Letter · Decision Letter 1]

14 Dec 2020

Dear Dr Xu,

We are pleased to inform you that your manuscript entitled "Vestigial mediates the effect of insulin signaling pathway on wing-morph switching in planthoppers" has been editorially accepted for publication in PLOS Genetics. Congratulations!

Yours sincerely,

Artyom Kopp

Associate Editor

PLOS Genetics

Bret Payseur

Section Editor: Evolution

PLOS Genetics

Comments from the reviewers (if applicable):

Reviewer's Responses to Questions

**Comments to the Authors:**

Reviewer #1: The authors have addressed most of my comments to my satisfaction.

The discussion of the developmental mechanisms underlying wing polyphenism in ants is only 1 sentence long . I was hoping for a more in depth comparison.

Reviewer #3: All points I suggested in the first review are sufficiently addressed.

Although it is very minor point, if authors prefer to precisely follow the Drosophila gene name, serrate and distalless should be Serrate and Distal-less, respectively, in the text and S2 Fig.

**Have all data underlying the figures and results presented in the manuscript been provided?**

Reviewer #1: Yes

Reviewer #3: None

PLOS authors have the option to publish the peer review history of their article (what does this mean?). If published, this will include your full peer review and any attached files.

Reviewer #1: No

Reviewer #3: **Yes: **Takahiro Ohde

**Data Deposition**

http://datadryad.org/submit?journalID=pgenetics&manu=PGENETICS-D-20-01558R1

**Press Queries**

---

## [Editor Report · Acceptance letter]

5 Feb 2021

PGENETICS-D-20-01558R1 

Vestigial mediates the effect of insulin signaling pathway on wing-morph switching in planthoppers 

Dear Dr Xu, 

We are pleased to inform you that your manuscript entitled "Vestigial mediates the effect of insulin signaling pathway on wing-morph switching in planthoppers" has been formally accepted for publication in PLOS Genetics! Your manuscript is now with our production department and you will be notified of the publication date in due course.

With kind regards,

Alice Ellingham

PLOS Genetics

On behalf of:
